# Impacts of Acute Sucralose and Glucose on Brain Activity during Food Decisions in Humans

**DOI:** 10.3390/nu12113283

**Published:** 2020-10-27

**Authors:** Xiaobei Zhang, Shan Luo, Sabrina Jones, Eustace Hsu, Kathleen A. Page, John R. Monterosso

**Affiliations:** 1Department of Psychology, University of Southern California, Los Angeles, CA 90089, USA; xiaobeiz@usc.edu (X.Z.); shanluo@usc.edu (S.L.); eustaceh@usc.edu (E.H.); 2Division of Endocrinology, Keck School of Medicine, University of Southern California, Los Angeles, CA 90033, USA; sljones@usc.edu (S.J.); katie.page@usc.edu (K.A.P.); 3Diabetes and Obesity Research Institute, Keck School of Medicine, University of Southern California, Los Angeles, CA 90033, USA; 4Neuroscience Graduate Program, University of Southern California, Los Angeles, CA 90089, USA

**Keywords:** glucose, sucralose, FMRI, food decision

## Abstract

It is not known how acute sucralose and glucose alter signaling within the brain when individuals make decisions about available food. Here we examine this using Food Bid Task in which participants bid on visually depicted food items, while simultaneously undergoing functional Magnetic Resonance Imaging. Twenty-eight participants completed three sessions after overnight fast, distinguished only by the consumption at the start of the session of 300 mL cherry flavored water with either 75 g glucose, 0.24 g sucralose, or no other ingredient. There was a marginally significant (*p* = 0.05) effect of condition on bids, with 13.0% lower bids after glucose and 16.6% lower bids after sucralose (both relative to water). Across conditions, greater activity within regions a priori linked to food cue reactivity predicted higher bids, as did greater activity within the medial orbitofrontal cortex and bilateral frontal pole. There was a significant attenuation within the a priori region of interest (ROI) after sucralose compared to water (*p* < 0.05). Activity after glucose did not differ significantly from either of the other conditions in the ROI, but an attenuation in signal was observed in the parietal cortex, relative to the water condition. Taken together, these data suggest attenuation of central nervous system (CNS) signaling associated with food valuation after glucose and sucralose.

## 1. Introduction

Glucose is the main circulating sugar in the blood. Both plasma glucose levels and insulin (which is released in response to rising glucose) have been shown to alter signaling in reward pathways at biologically relevant levels [1]. Using functional magnetic resonance imaging (fMRI) combined with a stepped hyperinsulinemic, euglycemic–hypoglycemic clamp technique, hypoglycemia (relative to euglycemia) is reliably associated with greater activation during food-cue exposure in the insula and striatum, along with higher hunger ratings. Circulating level of glucose has been found to modulate brain reward circuity (increased food-cue induced activity in the insula and striatum) and associated subjective reports of food motivation [2]. Related evidence linking insulin sensitivity to hunger and hunger signaling has been observed in studies utilizing oral glucose intake [3]. Food-cue responses in normal weight participants were attenuated after ingestion of glucose (dissolved in water) in the basal ganglia and paralimbic regions, as were ratings of subjective hunger [4]. 

However, Heni and colleagues [5] did not observe significant attenuation of food- cue- responsivity following glucose ingestion, suggesting the possibility that methodological variance (e.g., timing of intake relative to cue exposure) may be important.

Non-nutritive sweeteners (NNSs) are sugar substitutes which imbue foods with a sweet taste without adding calories or triggering a glycemic response [6]. The health consequences of chronic NNS consumption remains controversial [7,8,9,10,11,12,13]. It has been argued that NNS consumption disrupts learned responses that normally contribute to glucose and energy homeostasis and may drive metabolic dysregulation and increase the risk of obesity, diabetes, metabolic syndrome, and cardiovascular disease [9,13]. According to this hypothesis, by decoupling sweet taste receptor activity (normally a reliable cue of sugar consumption) from a subsequent rise in nutrient availability, learning processes that ordinarily support appetite regulation are disrupted [13,14]. However, an obesogenic impact of NNS has not been convincingly established. A review of studies investigating the relation of NNSs and weight gain and obesity concluded that the hypothesized link between NNS and obesity lacked empirical support [15]. Indeed, several well-designed intervention studies showed that chronic and covert NNS substitution could reduce energy intake and body weight [7,11]. A systematic review with meta-analysis also reported NNS use in place of sugar leads to reduced energy intake and reduced body weight [12]. The NNS sucralose, which is commonly used in the world food supply [16], appears to be particularly promising with regard to appetite suppression. A recent 12-week intervention study comparing the effects of NNSs and sucrose found decreased energy intake with sucralose consumption (but not with other NNSs) in overweight and obese individuals [8].

### Present Study

While the acute effects of sucralose consumption on appetite signaling may diverge with chronic sucralose ingestion effects, delineation of acute effects can provide clues regarding chronic effects. While some studies assessed the immediate gustatory responses and neural activity after consumption of sucralose [17,18], no study, to our knowledge, has directly examined the effect of acute sucralose intake on brain activity during food decisions. By including water, glucose, and sucralose conditions within the same study, we intended to distinguish acute effects of sweet taste (present in both sucralose and glucose conditions) from the effects of the caloric load (present only in the glucose condition).

We additionally considered whether response to very brief (subliminal) presentation of food- cues is sensitive to metabolic state [19]. Subliminal food primes have been shown to recruit activity within a distinct neural pathway [20]. Our intent in including subliminal primes was so that if we did observe study drink effects on appetite signaling during decision-making, the subliminal primes might allow us to address whether metabolic state affected the rapid reward-orienting response to stimuli [20,21], or was limited to the slower central nervous system (CNS) signaling associated with bidding on available food. However, perhaps due to the small number of subliminal presentations in this study compared to past reports, we did not observe any effect of subliminal presentations, even collapsing across all conditions. Therefore, we do not discuss the subliminal primes further (though for completeness, they are presented in Appendix B).

In the present neuroimaging study, participants bid money on visually depicted food. For each food, the participant’s bid could determine whether the food was available upon completion of the task. The task thus included both food-cue exposure, and food decision making. Some brain regions have consistently been found activated when participants view visual food-cue pictures, including the amygdala and hippocampus [22,23,24,25], striatum [22,23,26], orbital frontal cortex (OFC) [22,24,27], and insula [22,23,24,26,28]. We utilized a food-cue mask inclusive of these regions to investigate brain food-cue reactivity in our decision-making task. We examined the impact of study drinks on brain response during food decisions, and especially on what brain activity predicted the amount of money participants bid on particular food items. We hypothesized that relative to water, the consumption of glucose and possibly sucralose would attenuate the MRI signal in regions that track food valuation, and would lead to lower bids on food items. Identifying neural correlates of acute glucose and sucralose consumption during food choice may provide clues that inform the understanding of appetite regulation and obesity. Moreover, gaining a better understanding of underlying brain function and how this relates to motivated behaviors may also provide insights that extend to other disease processes with overlapping neural pathways, such as drug addiction [29].

## 2. Materials and Methods

### 2.1. Participants

Study recruitment was done primarily through flyers placed near the campus of the University of Southern California. Participants‘ characteristics are depicted in the result section. Due to excessive motion artifact or incomplete coverage, imaging data could not be included from one glucose session, one sucralose session, and two water sessions. To reduce variance across sessions related to hormonal change, female participants (with one exception) completed both scanning sessions in the window between 15 and 22 days post-start of their most recent menstruation (presumed luteal phase, though not confirmed by bioassay) [30,31]. Participants gave written informed consent to all experimental procedures approved by the Institutional Review Board of the University of Southern California. The study was conducted in accordance with the Declaration of Helsinki, and the protocol was approved by the Ethics Committee of the University of Southern California (UP-16-00413).

### 2.2. Stimuli

In each administration of the Food Bid Task, participants were presented with 30 visually depicted food items selected from “Food.pics”, a freely available database of 568 food images, each with associated normative data [32]. Normative data include rating scores (derived from approximately 2000 adults) of palatability, desire to eat, complexity, recognizability, and valence, as well as nutrient information for the item (mg of protein, fat and carbohydrates, kcal). The 30 items selected were judged by the research team to be easily and unambiguously identifiable to participants across a range of ages and backgrounds.

### 2.3. Procedures

Participants reported to Dana and David Dornsife Neuroimaging Center at USC between 8 and 9 am after an overnight fast. The start time for each of the three sessions, never varied for a participant by more than 30 min. Weight was measured at each session. MRI was performed using a 3-T Siemens MAGNETOM Tim/Trio scanner (Munich, Bavaria, Germany) with a 32-channel head-coil. See the details of MRI imaging parameters and neuroimaging preprocessing procedures in Appendix B. Prior to consumption of any drink, participants were trained on the procedures used in the study and provided session baseline information including ratings of mood and hunger. Next participants ingested 300 mL of water with a zero-calorie, mild cherry flavoring either (1) with no other ingredient (“Water”), (2) mixed with 0.24 g sucralose (“Sucralose”), or (3) mixed with 75 g glucose (“Glucose”). Drinks were prepared by the study coordinator. The rest of the study team and the participant were blind to drink type. The order of drinks used in the three sessions was approximately balanced (Water first for 10 participants, Glucose first for 10 participants, and Sucralose first for 8 participants). Participants were instructed to consume drinks in less than two minutes. Immediately after consumption, participants rated the pleasantness of the drink by rating scale (1–10). Testing on the Food Bid Task began approximately 55 min after consumption. Prior to the Food Bid Task, participants completed one or more fMRI tasks that had no connection to food, and are not discussed in this report. Sessions were no fewer than 2 days apart, and no greater than 30 days apart. In order to increase the valuation of depicted food items, study sessions continued for 30 min after completion of the scan, and participants were made aware that their only opportunity to eat would be if they bid enough (see below) on a randomly selected food item. Figure 1 provides the timeline of measures included in this report.

### 2.4. Appetite Rating

Participants reported on their appetite three times during each session: (1) at the start of the session (prior to consumption of the study drink), (2) just before entering the scanner (approximately 5 min after completing consumption of the study drink), and (3) upon completing the scanning session (approximately 90 min after completion of the study drink). For this report, we consider only responses to these questions, “How hungry do you feel right now?” and “How much do you want to eat something sweet?” Participants responded to these questions on a scale from 0 to 100 with 0 representing “not at all” and 100 representing “a lot.”

### 2.5. Food Bid Task

The food bid task was adopted from Suzuki et al. (2017) [33]. The task utilized the Becker–DeGroot–Marschak (BDM) auction method [34] to elicit participants’ valuation for food items. Participants were endowed with $5 that, if not spent, would be given as bonus compensation. In each trial of this task, the participant selected a bid ($0, $1, $3, or $5) indicating the amount that they were willing to pay for the depicted food. At the end of the experiment, the computer randomly selected one of the trials from the imaging session to be implemented (i.e., making that trial “real”, and the others not). If an item was randomly chosen, an associated “price” was randomly generated for it (either $0, $1, $3, or $5, with equal probability). If the participant’s bid was lower than the price, then they did not receive the food (and instead kept the full $5 endowment). If the participant’s bid was equal to or greater than the price, the price (not the bid) was deducted from their $5 endowment and the participant received the food item (plus any remaining amount from the $5). For example, if the bid amount for an item was $3, and the price for the item was $1, the participant would receive the food, and $4 (the amount remaining from the $5 endowment after paying the price). Participants went through several practice examples until comfortable with the procedure. The BDM auction method is incentive-compatible in the sense that the optimal strategy for participants is to always bid the amount that is closest to their true willingness to pay for the depicted item.

Within each imaging session, participants bid on each of the 30 food items during each of two task runs. Thus, in total, participants made a bid two times for each food item during each session. Prior to half of the trials the depicted food was first subliminally presented, but as noted above, we saw no evidence that this impacted brain activity or behavior even when all conditions were combined, and so investigating differences between conditions was not warranted (though is included in Appendix B for completeness). The general timeline for each trial is presented in Figure 2. A blank white screen with fixation cross was presented for a jittered duration, with a mean of 3.5 s and exponential distribution. This was followed by supraliminal presentation of the food for that trial which remained visible for 3 s. Next the participant indicated a bid on the item, within 3 additional seconds, by pressing the key on a keypad that corresponded to the intended dollar amount. Importantly, although participants could not enter bids until after the visual food-cue disappeared, the requirement to bid was predictable and so it is likely that participants were formulating their bids while the food was visually presented. Therefore, we did not separate these periods with a jitter in the design phase (they occur in temporal lock-step) and instead treated the period beginning with cue-presentation and ending with bid entry as a general “food valuation period” in analyses. Mappings between keys and bid amounts were randomized across trials in order to dissociate the bid amount from the spatial information. The bid the participant made was visually presented in the center of the screen immediately after the participant’s keypress (feedback phase, 0.5 s). At the end of each trial, a blank white screen with fixation cross was presented during an intertrial interval (ITI phase), again with duration jittered using an exponential distribution with a mean of 3.5 s.

### 2.6. Attribute-Rating Task (Outside the MRI Scanner)

Following the procedure of Suzuki et al. (2017), after all three imaging sessions were completed, participants estimated the nutrient content for each depicted food item. See Appendix B for details. Participants were not informed that they would be providing these estimates prior to the task. In addition, participants were asked to report their guess for the market price for each food item. The order of the questions was randomized across participants.

### 2.7. Data Analyses

Primary analyses used linear mixed-effects models (LMMs; [35]). This approach allowed us to take the full response patterns into account, without averaging over individual items or conditions. F and *p* values were obtained using the lmerTest package [36], which uses the Satterthwaite approximation for degrees of freedom. Pairwise comparison using Holm–Bonferroni procedure adjustment was done using the emmeans function in the emmeans package [37] when significant effects were shown. Repeated measures ANOVA (Analysis of variance) was used when analyzing the appetite scores.

#### 2.7.1. Appetite Analysis

Before analyzing the appetite for the sweet food and hunger score, we first replaced the 38 missing values (7.5% of the scores were missing/incomplete and missing). Since we had no reason to expect a relationship between the study manipulation and missing data, missing data were imputed using the predictive mean matching (pmm) method from the mice package in R [38], taking the averaged imputed value over 50 imputations. However, we also carried out analyses using the data without imputation. Both one-way and two-way repeated measures ANOVA were used to test the baseline appetite scores difference and measurement time and drink effects on Appetite score changes from baseline. Drink type, measurement time, and the interaction between measurement time and drink type were the fixed-effects terms. Drink type was also a random slope nested within a random intercept participant term, taking into account intra-individual variability.

#### 2.7.2. BID Task Data Analysis

In our primary LMM analysis of bids during the Food Bid Task, participants’ bids for the food items were our dependent variable, and the fixed effect was drink type (water, sucralose, or glucose) and the random effects included random intercept and slope for the drink type by participant. Baseline hunger and appetite for sweet food were included as covariates in order to minimize the impact of pre-study drink session variance in appetite (e.g., a participant that happened to be hungrier when arriving at their water session than their glucose session). BMI and gender were additionally included as covariates since past research has shown that BMI and gender could be moderators of cue reactivity [39,40].

#### 2.7.3. Neuroimaging Data Analyses

Data were processed using the fMRI Expert Analysis Tool (FEAT) version 6.0. In addition to six motion parameters, we included nuisance regressors for time points corresponding to motion outliers for both models using the FSL motion outliers program (http://fsl.fmrib.ox.ac.uk/fsl/fslwiki/FSLMotionOutliers), which defined outlier time points using the upper threshold of the 75th percentile plus 1.5 times the interquartile range. Temporal derivatives and temporal filtering were added to increase statistical sensitivity. Inter-trial interval periods were not modeled, and therefore provided the implicit baseline for analyses. For imaging analyses, we were primarily interested in the effects of study drink on (1) overall activity during food valuation, and (2) activity tracking participant bids during food valuation.

Our analyses focused on the food valuation period, which began with the presentation of the food picture for the trial and ended when a bid was recorded (see Figure 2). The first general linear model (GLM) included (1) food valuation period unweighted and (2) food valuation period weighted by bid for the food. Trials in which participants did not respond were modeled with a separate regressor of no interest, as were the regressors for subliminal priming (see Appendix B).

##### Region of Interest (ROI) Analyses

ROI analyses were carried out in order to examine several possible effects of the manipulation. Below we describe these ROIs, which were directed at (1) examining response to depicted food stimuli within brain areas known to be active during visual food-cues (food-cue ROI, see Appendix B for details), and (2) examining functional connectivity with the portion of the medial orbital frontal cortex (mOFC) (previously implicated in value-tracking) that overlapped with bid-related activity in our analysis (mOFC ROI).

***mOFC ROI*:** In order to examine study manipulation effects on value-tracking activity, we focused on the medial orbitofrontal cortex, based on the extensive literature establishing its importance both in the context of food valuation [33,41] and of valuation more generally [42,43,44]. Because the medial OFC is large and heterogeneous, we limited the ROI to the overlap between the anatomically defined medial OFC based on the AAL database [45] and the cluster-map identifying bid-tracking activations during food valuation (across conditions to avoid “double dipping” confound, see [46]). An exploratory psychophysiological interaction (PPI) analysis was then performed in which we used the time-series of mean activity in this cluster as a seed to predict activity throughout the rest of the brain during the food valuation period vs. rest. Group-level statistics images were thresholded with a cluster-forming threshold of z > 3.1 and a Bonferroni corrected cluster probability of *p* < 0.05. Three group-level paired-t analyses (sucralose vs. water, glucose vs. water and sucralose vs. glucose) were performed in FEAT using a mixed-effects model, with FSL’s FLAME1 option with outlier deweighting.

Across different ROI analyses, individual signal changes (beta values from statistical models) were extracted separately for each participant during each session. LMM analysis was done to test the drink effect on brain signal change for each of the ROIs described above with drink type as a fixed effect and participant entered as a random intercept with baseline hunger, appetite for sweet food, BMI and gender as covariates.

##### Whole-Brain Analyses

In addition to ROI analyses, whole brain analyses were carried out. Group analysis with multiple sessions for each subject was performed in FEAT using a mixed-effects model, with FSL’s FLAME1 option. In accordance with the model discussed above, group-level paired-t analyses were carried out (sucralose vs. water, glucose vs. water and sucralose vs. glucose) with FEAT using a mixed-effects model. For these we utilized FSL’s FLAME1 option with outlier deweighting. All statistical maps were cluster corrected for multiple comparisons (cluster height threshold: Z > 3.1; cluster significance: *p* < 0.05).

## 3. Results

### 3.1. Participants Characteristics

Participants were right-handed (in order to reduce heterogeneity in the mapping of function to brain area, which can differ in left-handed individuals), nonsmokers, weight stable for at least 3 months, non-dieters, not on any medication (except oral contraceptives), with normal or corrected- to-normal vision, and no history of diabetes, eating disorders, or other significant medical diagnoses. Twenty-eight volunteers (14 females, mean age = 25.36 ± 4.74, range 19–36 years old) with no history of eating disorders, diabetes, or other major medical illnesses participated in the study, see characteristics of subjects included in the final analyses in Table 1. Seven participants (25%) had a body mass index (BMI) index in the normal weight range (18.5 to <25 kg/m^2^), 14 participants (50%) had BMI in the overweight range (25 to <30 kg/m^2^) and 7 participants (25%) had BMI in the obese range (>=30 kg/m^2^) (classification based on World Health Organization criteria [47]) All 28 participants completed two runs of the Food Bid Task on each of three days (with the exception of one run for one participant on one day due to a time constraint).

### 3.2. Study Drink Rating

Participants rated both glucose (t(54) = 2.69, *p* = 0.03) and sucralose (t(54) = 2.62, *p* = 0.03) as more pleasant than water. Participants also rated both glucose (t(54) = 15.65, *p* < 0.0001) and sucralose (t(54) = 14.96, *p* < 0.0001) as sweeter than water. Participants reported similar ratings of the pleasantness (t(54) = 0.07, *p* = 0.94) and sweetness (t(54) = 0.69, *p* = 0.49) of the glucose and sucralose drinks.

### 3.3. Appetite Rating

There were no baseline difference in hunger (F(2,54) = 0.13, *p* = 0.88) or appetite for sweet food (F(2,54) = 1.27, *p* = 0.29)) prior to drink consumption. In addition to pre-drink appetite ratings, as noted above, post-drink appetite ratings were acquired approximately 5 min after drink consumption, and then again approximately 90 min after drink consumption. Post drink appetite ratings were analyzed as difference scores relative to pre-drink ratings. Two by three repeated measures ANOVAs were carried out to examine the effects of measurement time (appetite change at first and second post-drink assessment points), and drink effects (glucose, sucralose and water drinks) on changes from baseline.

For hunger score (Figure 3a), a significant main effect of measurement time (F (1,27) = 50.02, *p* < 0.001), a marginally significant main effect of drink (F (2,54) = 2.80, *p* = 0.07) and a significant interaction effect between measurement time and drink (F (2,54) = 4.39, *p* = 0.02) were found. The first hunger score change (approximately 5 min after drink consumption) did not differ significantly between drinks (all *p* values of paired t comparisons with Holm method multiple comparison correction were larger than 0.1). The second hunger score (approximately 90 min after drink consumption and after completing the Food Bid Task) increased less after glucose than after sucralose (t(88.7) = −3.078, *p* < 0.01) or water (t(87.5) = −3.084, *p* < 0.01) and no hunger difference was found between water and sucralose consumption (t(88.7) = 0.28, *p* = 0.78). For appetite for sweet food (Figure 3b), only a significant main effect of measurement time (F (1,27) = 12.95, *p* = 0.001) was observed, with the second sweet food craving score change significantly higher than the first change.

### 3.4. Food Bid Task

The Food Bid Task began approximately 55 min after consumption of the study drink. Across all participants, 4.3% of the trials (221/5110) were excluded from the task analysis due to the absence of responses within the allotted time. There was a marginally significant (trend) drink effect on participant bids for depicted food (F (2, 19.73) = 3.39, *p* = 0.054). Post-hoc comparisons indicated that this overall marginally significant trend was driven by a marginally significant trend towards lower bids after glucose relative to water (z = −2.25, *p* = 0.07) as well as after sucralose relative to water (z = −2.28, *p* = 0.07). No significant difference was found between sucralose and glucose (z = 0.018, *p* = 0.99), each with Holm–Bonferroni adjustment for multiple comparisons. The mean bid in the water condition was $1.93, in the glucose condition was $1.68 (13.0% lower than water condition), and in the sucralose condition was $1.61 (16.6% lower than water condition). The difference in bids for each food item between different drink conditions are presented in Table 2. No effect of gender (F(1,18.97) = 0.71, *p* = 0.41) and BMI (F(1,20.49) = 0.71, *p* = 0.12) was observed.

In a secondary (post hoc) analysis, we removed items for which participants bid “0” on all the six experimental runs (3 drinks, with 2 runs per drink), treating these as missing values. We reasoned that these items may have had no value (perhaps the participant did not like or want to eat the food even if free) and so could have reduced power to detect changes between conditions. With these items removed (mean of 3.0 out of 30 food items per participant) the drink effect on bids was significant overall (F (2, 20.29) = 3.70, *p* = 0.04). No effect of gender (F(1,19.37) = 0.80, *p* = 0.38) or BMI (F(1,20.25) = 2.26, *p* = 0.15) was observed.

Finally, we carried out correlational analyses directed at identifying attributes (based on available normative data of perceivers’ judgments for the stimulus set) of the 30 food pictures that were associated with how glucose and sucralose impacted bids. Although not corrected for multiple comparisons, there was some evidence that glucose (relative to water) led to reduced bids more for foods high in sugar, high in calories, high in fat. While a similar pattern was observed with regard to sucralose, correlation coefficients were generally lower and non-significant (see Table 3).

### 3.5. The Effect of Study Drink on Food-Cue Network ROI Activity during Food Valuation

Beta values were extracted from the a priori food-cue ROI [29] discussed above (all clusters combined). A significant drink effect was found for beta-values within the network during food valuation (F(2,32.20) = 4.52, *p* = 0.02). In post-hoc comparisons, beta-values were observed to be significantly lower after sucralose relative to water (t (23.3) = −2.91, *p* = 0.02). Beta-values did not differ significantly between glucose and water (t(22.8) = 1.70, *p* = 0.20) nor between glucose and sucralose (t(24.1) = 1.64, *p* = 0.20). Pre-drink appetite for sweet food (which was included as a covariate in the model) significantly predicted beta values overall, with increased signal when participants had higher baseline appetite score for sweet food (β = 0.24, *p* = 0.03). No significant effect of BMI or Gender was found. See Figure 4 for mean beta-values within the food-cue mask in each of the drink conditions.

In order to allow visual comparison of the main effect of glucose and sucralose (each relative to water) on primary dependent variables (bids and brain signal within the food-cue ROI) along with appetite scores, we first normalized all dependent variables as z-scores, and then plotted the 95% confidence intervals for all comparisons. These are presented in Figure 5, with order matching the temporal sequence of the acquisition of the measure. Each of these main effects is described above, but the conversion to z-scores and the ordering in temporal sequence allows the main effects to be better visualized. The only significant difference between glucose and sucralose was that the second hunger score (approximately 90 min after drink consumption) increased less after glucose than after sucralose (t (88.7) = −3.078, *p* < 0.01).

### 3.6. Whole Brain Analyses of Activity during Food Valuation Period

Overall increase in brain activity during food valuation across drinks was, as expected, quite extensive (see Figure 6 below). It included the network of regions previously linked to food-cues: lateral OFC, dorsolateral prefrontal cortex (dlPFC), left ventral striatum, left amygdala, hippocampus, bilateral anterior insula, bilateral middle insula, bilateral precuneus, and left postcentral gyrus), as well as the fronto-parietal network which is generally active during decision- making tasks [48,49,50], and in the visual cortex.

### 3.7. Drink Effects on Whole Brain Activity Associated with Food Valuation

In secondary whole brain analyses, we compared neural activity during food valuation for each drink condition (glucose vs. water, sucralose vs. water, and glucose vs. sucralose). The only cluster in which a significant activity difference between the glucose and water condition was present was a cluster in the left parietal lobe (partially overlapping the postcentral gyrus) in which activity after glucose was significantly diminished relative to water. This cluster overlapped the network of regions that were generally active during food valuation (see Figure A2a in Appendix B). For the comparison of sucralose and water, significantly lower activity was observed after sucralose in a set of regions that included left dlPFC, visual cortex, frontal gyrus, and cingulate (mostly posterior), precuneous, supplementary motor cortex, frontal operculum, each of which overlapped with the areas showing general increase in activity during food valuation (see Figure A2b in Appendix B). No significant differences were observed in the comparison between glucose and sucralose conditions.

### 3.8. Bid-Correlated Brain Activity during Food Valuation Period

Regions in which brain activity during food valuation was positively associated with bids across drinks are presented in Figure 7a.

Many of the regions associated with bid (greater signal on trials in which participants bid more money) overlapped the food valuation period main effect contrast map (Figure 7a). This included bid-correlated clusters in the OFC, visual cortex, cingulate, paracingulate gyrus, frontal pole, frontal gyrus, thalamus, caudate, brain stem, hippocampus, putamen, and accumbens. Exceptions to this overlap with the general food valuation activity map were a large cluster within the medial OFC and small bilateral clusters in the frontal pole, which tracked food bids but which were not generally active during food valuation relative to rest. No significant differences were observed between drink sessions on the association between bids and brain activity.

Because the medial OFC has been implicated in valuation, and was prominent in our map of bid tracking, we carried out an exploratory PPI in which we (1) identified clusters in which activity was more correlated with the seed during food valuation, and (2) investigated whether there were regions in which connectivity with the bid-tracking seed significantly differed based on drink condition. For example, it might be the case that regions preferentially sensitive to particular qualities of depicted food (e.g., sweetness) would differ in their association with the medial OFC seed as a function of drink. As shown in Figure 7b, we identified regions more correlated with the orbitofrontal seed during the food valuation period than in other periods of the task. Significant functional connectivity with the medial OFC was observed in a bilateral network of regions that included the caudate, anterior insula, and nucleus accumbens, as well as part of the frontal pole, part of lateral OFC. In contrasts of PPI results between drink conditions, no significant differences were observed either in the whole brain nor in the areas showing positive functional connectivity.

## 4. Discussion

Our study aimed at investigating the effects of acute ingestion of glucose and of sucralose on subjective hunger, food valuation, and associated brain activity. In interpreting our results, it is important to keep in mind both the baseline metabolic state (overnight abstinence) and the timing of assessments relative to consumption.

### 4.1. Primary Findings Related to Acute Glucose Consumption

Relative to water, glucose intake was associated with (1) significantly reduced change in subjective hunger 90 min after consumption (t(87.5) = −3.08, *p* < 0.01) but not 5 min after consumption (t(87.5) = 0.74, *p* = 1)), marginally significant reduction in monetary bids on foods approximately 55 min after consumption (z = −2.25, *p* = 0.07), and reduced signal change during food valuation within a small cluster of the left parietal cortex (which was part of the extensive map in which signal was elevated during food valuation with all conditions combined). Within the food-cue ROI, signal change was not significantly attenuated after glucose (t (22.8) = −1.70, *p* = 0.20) though the pattern of results was in the expected direction.

The general pattern of findings in the glucose consumption condition is consistent with prior research. One study that directly examined the food-cue reactivity after glucose ingestion revealed decreased in-scanner food-cue induced hunger and desire for food scores compared with water [51]. Participants in that study also indicated lower amounts of money that they were willing to pay for food after glucose ingestion. Moreover, a meta-analysis showed that low level of blood glucose increases the willingness to pay for food [52]. Evidence has shown that ingestion of energy containing glucose elicits a decrease in food-cue reactivity in brain regions associated with hunger [18,53,54]. While the area within the left parietal lobe in which reduced activity post-glucose (relative to water) was observed is not implicated in appetite signaling, it has been implicated in visual attention orientation [55] and food reward during visual food-cue presentation [56]. Indeed, attention related activity in the parietal lobe in response to food stimuli has been reported to be greater among individuals with higher BMI. Therefore, the observed attenuation after glucose consumption could reflect a decrease in attention to food-cues.

### 4.2. Primary Findings Related to Acute Sucralose Consumption

Relative to water, sucralose intake was associated with marginally significant reduction in monetary bids on foods approximately 55 min after consumption (z = −2.28, *p* = 0.07), and reduced activity within the a priori food-cue ROI (t(23.3) = −2.91, *p* = 0.02) and several additional regions including the visual cortex, dorsolateral prefrontal cortex, and posterior cingulate. Collectively, the findings provide evidence that at approximately 1-h post consumption, sucralose reduces CNS activity associated with food valuation. This acute appetite suppression effect may contribute to aforementioned recent evidence that consumption of sucralose sweetened beverages can result in decreased energy intake throughout the 12-week intervention and reduced BMI after the intervention (compared with baseline) in overweight and obese individuals [8].

Although the mechanism of the observed diminished food-cue ROI activity after acute sucralose consumption is not clear, it is likely that receptors in the mouth or gut that are normally sensitive to sugars play a role. Sweet taste perception of both sugars and NNSs is peripherally mediated by the T1R3 and T1R2 receptors on the tongue [57], though only the T1R3 appears to be sensitive to sucralose [58]. Sucralose has high binding affinity (lower dissociation constant) to T1R3 receptor compared to glucose [59]. The upper gut has receptors that respond to sweetness, leading to satiety hormone release. It has been found both sucralose and glucose could induce GLP-1 release in a human L cell line [60]. A recent study [61] examined the function of the sweet taste receptor in arcuate nucleus using sucralose and the findings suggested that sweet taste receptors could lead to anorexigenic signals in the brain and thus reduce food intake. Though NNSs were initially considered to be without glycemic [6] and metabolic effects, more recent evidence suggests that NNSs may have metabolic effects [9].

### 4.3. Comparison of Acute Sucralose vs. Acute Glucose Consumption

Contrary to our expectation that glucose consumption relative to sucralose consumption would cause greater reduction in food bids and attenuation in brain activity associated with food decision making, no significant differences were observed. It is possible that differences would have been observed had the Food Bid Task been administered at a different time point. It is worth noting that appetite ratings, were not different between these conditions five minutes after the study drink consumption but were significantly lower after glucose 90 min after study drink consumption (see timeline in Figure 1).

### 4.4. Food Decisions and the Orbitofrontal Cortex

As expected, activity correlated with participants’ bids was observed throughout regions previously linked to appetite but was especially prominent in a large cluster of the medial orbitofrontal cortex. The mOFC was not part of the network that was generally recruited during the task, but its association with value is in keeping with an extensive literature in neuroeconomics [33,41,42,43,44]. Based on psychophysiological interaction analysis (PPI) we identified the mOFC cluster to be functionally connected during food valuation to a network of regions that included bilateral caudate, anterior insula, nucleus accumbens, the frontal pole, and part of lateral OFC. We observed no statistical evidence that drink consumption altered functional connectivity with the mOFC.

### 4.5. Limitation

Several limitations of this study are worth noting. First and perhaps most significantly, our sample size is relatively small. Small sample studies can lead to inconsistency, and serious problems of low replicability of neuroimaging results [62]. We have made our data accessible using the standardized “Brain Imaging Data Structure” (BIDS) format [63] to facilitate future meta-analyses that may more definitively address this issue. Second, our sample was quite heterogeneous in BMI [64]. While this heterogeneity might be a strength in a large study where associations with these variables could be explored, the modest sample size here does not provide sufficient power to do this. The heterogeneity may have resulted in loss of power to detect effects that would be evident in a more homogenous sample. A third limitation of our experimental design is that we did not record subjective hunger (appetite scale) for the 75-min window in which participants were completing the neuroimaging portions of the protocol. Given the dynamic nature of post-ingestive signaling relevant to hunger, regular assessment of appetite during this period would be informative.

## 5. Conclusions

We observed both behavioral and neural evidence of impact from acute glucose and acute sucralose on food decision-making. Most noteworthy, during the Food Bid Task (approximately 55 min after consumption of sucralose) brain activity within regions previously established as responsive to food-cues was attenuated, and there was a marginally significant reduction in participants’ bids on food items. Although the relationship between these observed acute effects and the effects of regular sucralose consumption are not clear, it is possible that acute effects could result in decreased energy intake over time. Future work should focus on investigating this possible connection, and on identifying the signaling mechanism underlying the observed effects.

## Figures and Tables

**Figure 1 nutrients-12-03283-f001:**
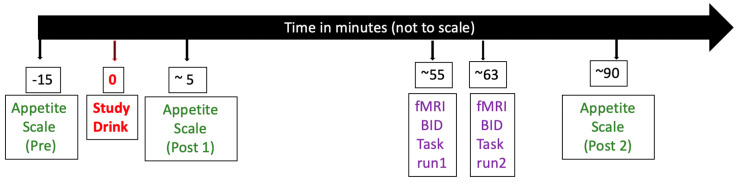
Timeline of the study. We use “~” sign as the approximation of the time points and FMRI BID task refers to functional magnetic resonance imaging food bid task.

**Figure 2 nutrients-12-03283-f002:**
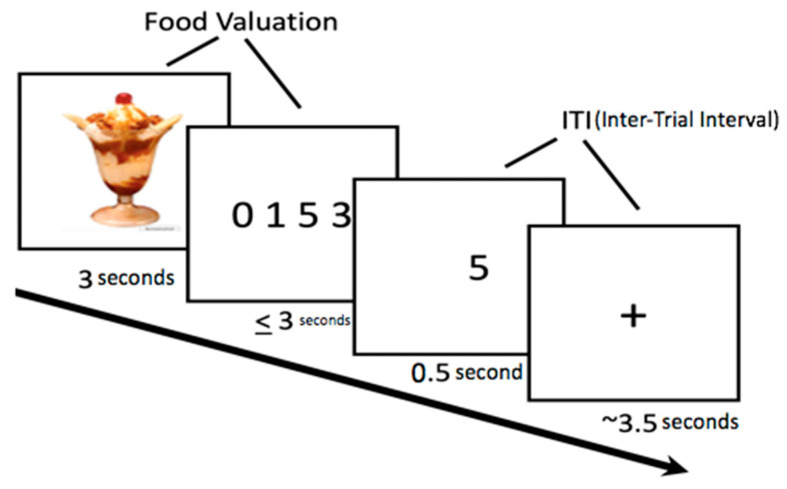
Food Bid Task trial structure.

**Figure 3 nutrients-12-03283-f003:**
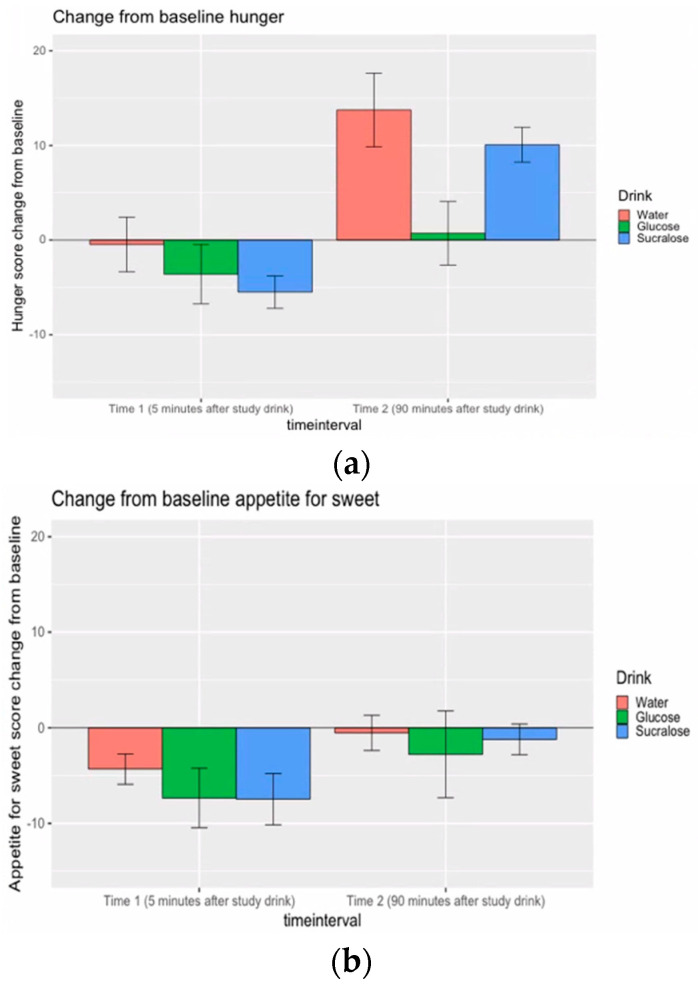
(**a**) Hunger score changes from baseline; (**b**) Appetite for sweet scores change from baseline.

**Figure 4 nutrients-12-03283-f004:**
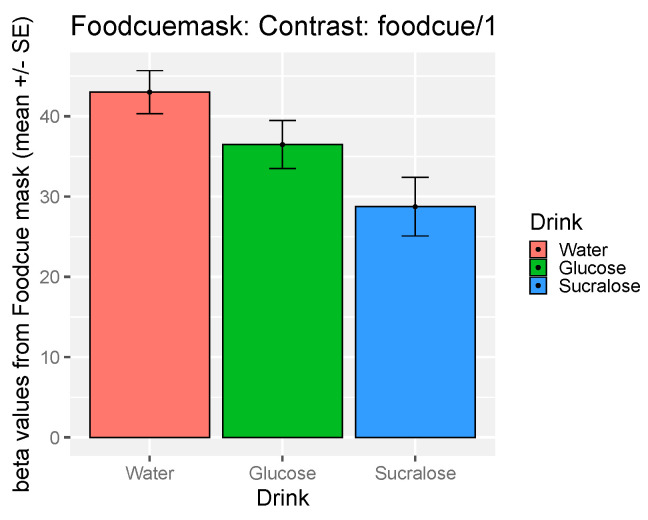
Signals change during food valuation in food-cue region of interest (ROI) by drinks.

**Figure 5 nutrients-12-03283-f005:**
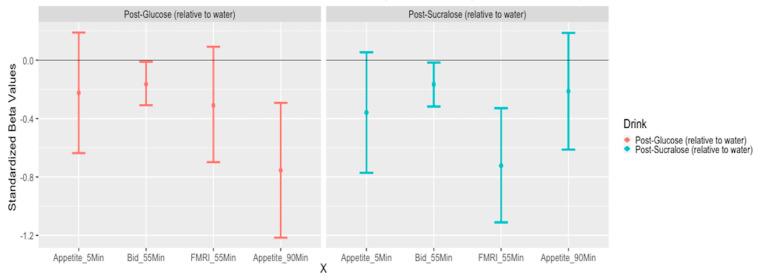
95% confidence intervals for the effect of Glucose and Sucralose (relative to water) on standardized (z-score) dependent variables. Four bars in each drink condition refer to hunger score about 5 min after drink ingestion, willingness to pay for the food about 55 min after drink ingestion, brain signals from food-cue a priori mask when viewing the food pictures about 55 min after the drink, and hunger score about 90 min after the drink ingestion. In general, intervals that do not intersect 0 indicate *p* < 0.05 for comparison with water condition. However, these confidence intervals do not reflect Holm–Bonferroni adjustment used in analysis of bids.

**Figure 6 nutrients-12-03283-f006:**
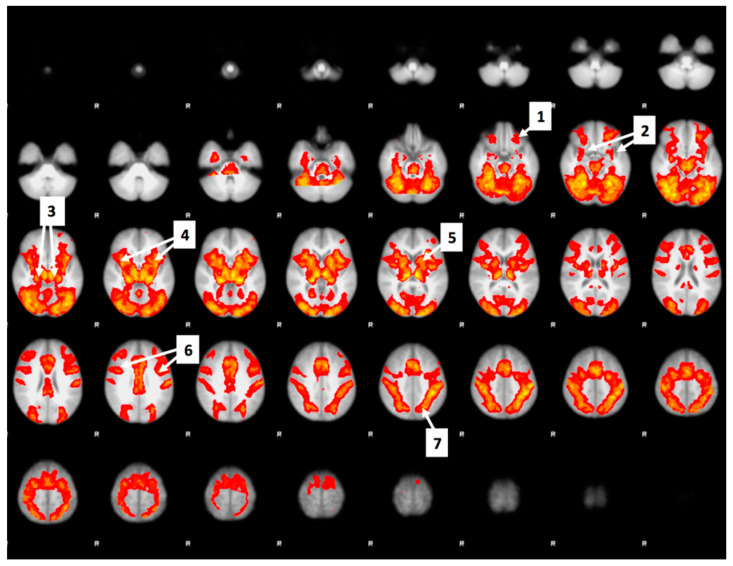
The numbered arrows in the figure point to the regions with increased signal during food valuation, which included: (1) lateral orbital frontal cortex (OFC); (2) amygdala; (3) hippocampus; (4) insula; (5) left ventral striatum; (6) postcentral gyrus; (7) precuneus.

**Figure 7 nutrients-12-03283-f007:**
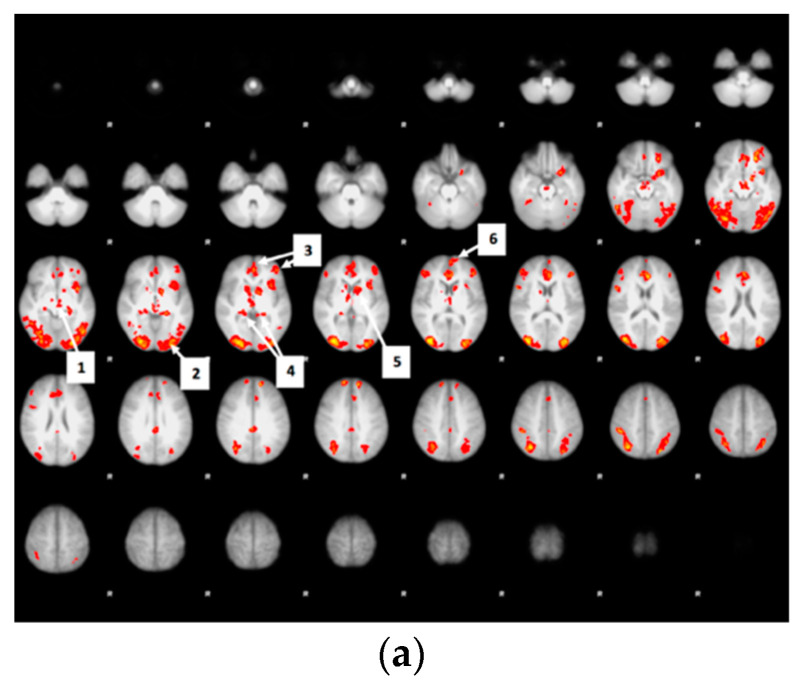
Schemes follow the same formatting. If there are multiple panels, they should be listed as: (**a**) The numbered arrows in the figure point to the brain regions in which activity was positively associated with bids, which included: (1) thalamus; (2) visual cortex; (3) medial and lateral OFC; (4) hippocampus; (5) caudate; (6) frontal pole.; (**b**) The numbered arrows in the figure point to brain regions that have positive connectivity with the mOFC during Food Valuation, which included: (1) lateral OFC; (2) frontal pole; (3) caudate; (4) insula.

**Table 1 nutrients-12-03283-t001:** Characteristics of subjects included in the final analyses.

Characteristic	Mean ± SD or *N* (%)
Gender	
Male	14 (50%)
Female	14 (50%)
Age (years)	25.36 ± 4.74 ^1^
BMI (kg/m^2^)	27.61 ± 5.02 ^1^
Ethnicity	
Caucasian	6 (21%)
Black or African American	7 (25%)
Hispanic or Latino	4 (14%)
Asian	10 (36%)
Other	1 (4%)
Education (degree)	
Bachelor’s	18 (64%)
Graduate school level	9 (32%)
High school	1 (4%)

^1^ Values are means± SDs (standard deviations). BMI—body mass index

**Table 2 nutrients-12-03283-t002:** Bid difference in cents between drinks.

		(Bid Difference from Overall Mean)		
Food Item	Mean Bid	Water	Sucralose	Glucose
Sundae	$2.33	+$0.56	−$0.24	−$0.33
Filled Chocolates	$1.73	+$0.51	−$0.13	−$0.39
Cheese and Cold Meat Platter	$1.73	+$0.31	−$0.28	−$0.02
Apple	$1.36	+$0.30	−$0.27	−$0.04
Waffle with Whipped Cream	$2.23	+$0.39	−$0.16	−$0.23
Sushi	$2.05	+$0.32	−$0.20	−$0.13
Tomatoes	$0.84	+$0.25	−$0.25	−$0.01
French Fries	$2.33	+$0.34	−$0.14	−$0.20
Gummi Candy and Licorice Mix	$1.15	+$0.27	−$0.20	−$0.07
Bowl of Rice	$1.16	+$0.28	−$0.18	−$0.11
Pizza (With Mushrooms)	$3.22	+$0.34	−$0.12	−$0.22
Crackers	$1.10	+$0.24	−$0.17	−$0.06
Nuts (Cashews)	$1.76	+$0.20	−$0.19	−$0.02
Cheese Platter	$1.84	+$0.21	−$0.17	−$0.05
Roast Beef	$3.03	+$0.15	−$0.22	+$0.06
Pizza (With Salami)	$3.33	+$0.24	−$0.12	−$0.11
Doughnut / Donut Jam	$1.84	+$0.25	−$0.06	−$0.20
Salad Plate	$2.02	+$0.13	−$0.17	+$0.04
Loaf of Bread	$1.29	+$0.06	−$0.20	+$0.13
Popcorn	$1.46	+$0.17	−$0.08	−$0.09
Crisp Bread	$0.80	+$0.08	−$0.16	+$0.07
Chocolate Muffin	$1.75	+$0.10	−$0.12	+$0.01
Broccoli	$0.92	+$0.16	$0.01	−$0.17
Peanuts	$1.04	+$0.02	−$0.07	+$0.06
Strawberries	$2.83	−$0.01	−$0.08	+$0.10
Banana	$1.19	−$0.01	−$0.07	+$0.07
Toast	$1.33	−$0.05	−$0.11	+$0.17
Opened Chips Bag	$1.44	+$0.09	+$0.04	−$0.12
Croissants	$2.28	−$0.10	+$0.03	+$0.06
Green Asparagus	$0.82	−$0.09	+$0.16	−$0.06

**Table 3 nutrients-12-03283-t003:** Association between Bid difference of drinks and food attributes.

Food Attribute	Water Bid– Sucralose Bid	Water Bid– Glucose Bid	Sucralose Bid– Glucose Bid
palatability	0.12	0.21	0.18
healthiness	−0.28	−0.41	−0.3 (*p* = 0.1095)
familiarity	−0.03	0.01	0.07
fat	0.26	0.4 ^2^ (*p* = 0.0275)	0.31 ^1^ (*p* = 0.097)
vitamin	−0.24	−0.33 ^1^ (*p* = 0.079)	−0.22
sodium	0.03	0.07	0.08
calorie	0.35 ^1^ (*p* = 0.058)	0.44 ^2^ (*p* = 0.015)	0.25
carb	0	0.1	0.18
sugar	0.3 (*p* = 0.113)	0.38 ^2^ (*p* = 0.0385)	0.23
protein	0.12	−0.06	−0.27

^1^*p* value < 0.1; ^2^
*p* value < 0.05.

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
