# Peer review of "Impacts of Acute Sucralose and Glucose on Brain Activity during Food Decisions in Humans"

_nutrients, 2020, doi:10.3390/nu12113283_

Round 1

Reviewer 1 Report

The study by Zhang et al. investigated the impact of acute sucralose and glucose on brain activity during food decisions in humans. Sucralose and glucose are common source of sweetness. In this study, the authors conducted Food Bid Task to how these two different sweeteners alter signaling within the brain when individuals make decisions about food. The authors provide solid evidence suggesting that glucose and sucralose lead to lower bids comparing to water. Interestingly, they found that sucralose, by not glucose, attenuates activity in priori ROI by fMRI. It would be informative if the authors can discuss the different signaling that may occur in brain between glucose and artificial sweeteners during making food decisions.

Reviewer 2 Report

The authors tried to analyze the impact of sucralose and glucose on brain activity during food decisions in human.This manuscript is clearly comprehensive and easy to ready, although it is a difficult topic. It is a interesting paper for many readers.  Results of this study (acute suppression of appetite by sucralose) may indicate that sucralose comsumption causes to decrease body weight through appetite suppression.

there is only minor concerns.

Table 1 may be included in results section.

Reviewer 3 Report

Nutrients_review                                                                                             Sep 18, 2020

The article “Impact of acute sucralose and glucose on the brain activity during food decisions in humans”, is an interesting and generally well-written article.  Having said this I have some concerns:

-the article may not be appropriate for our readership, and perhaps should be re-submitted for a neuro-imaging audience

-the methods is 5+ pages, very long, perhaps some of it could be submitted as supplemental

-authors need to provide the clinical context for the work (i.e is this meant for nutritionists, dieticians, general physicians, endocrinologists, athletes/trainers, diabetes researchers…?)

-there is quite a bit of quasi-and insignificant data described and presented, is this necessary?

-the article is verbose, taking the reader on an odyssey, authors would be encouraged to include essential information necessary for general readers to understand the content and findings

-the article need to be firmly framed for the readership of Nutrients

Specifically:

-the article characterized data on 28 patients, this could be included in the title (N=28), which would help to contextualize the finding from the beginning for the reader, hence the necessity for the stating a small sample size in the limitations rendered somewhat obsolete.

Line 87-89, hypothesis, what does this mean for a wider audience?

Line 93, participants are right-handed, why is this important?

Table 1 is oddly formatted, table columns should be listed as variable and outcomes, as it currently reads table headings are gender and male/female…& bolded, also data is formatted horizontally, normally data is formatted vertically?

Lines 199-202 were already described in Lines 123-126, could this be consolidated.

Line 220 and 265 GLM was already abbreviated, and acronym provided, check throughout text for consistency.

Line 266-267 awkward, consider revising.

Line 353-355 describing the secondary analysis, is this necessary, could this be framed differently, how is this relevant to the final outcomes.

Line 378, check spacing and orthography.

Figure 5, is the figure title meant to be the figure description?

All of the MRI figures could be supported by arrows of changing signal, to provide some sort of legend or explanation, this would help to link information for readers with out explicit granular knowledge on the brain. 

Line 428-432, should this have been explained in more detail (perhaps w/ a diagram) in the introduction, so that the reader has a greater understanding of where the authors focus is.

Line 479-481, again reporting statistically significant findings?  Should marginally significant findings be discussed…do they still have clinical implications, if so this should be framed in that way.

Line 484-486, highlighted, on purpose?  Please amend.

Round 2

Reviewer 3 Report

It appears as though the manuscript has been improved, however, may be out of scope for Nutrients.